# Safety and Immunogenicity of a Booster Vaccination by CoronaVac or BNT162b2 in Previously Two-Dose Inactivated Virus Vaccinated Individuals with Negative Neutralizing Antibody

**DOI:** 10.3390/vaccines10040556

**Published:** 2022-04-03

**Authors:** Kristi Tsz-Wan Lai, Emilie Yuen-Ting Lai Wan Loong, Terry Ling-Hiu Fung, Luke Wing-Pan Luk, Chor-Chiu Lau, Jonpaul Sze-Tsing Zee, Edmond Shiu-Kwan Ma, Bone Siu-Fai Tang

**Affiliations:** 1Department of Pathology, Hong Kong Sanatorium and Hospital, Hong Kong Special Administrative Region, Hong Kong 999077, China; kristi.tw.lai@hksh.com (K.T.-W.L.); htyemilie@gmail.com (E.Y.-T.L.W.L.); jonpaul.st.zee@hksh.com (J.S.-T.Z.); eskma@hksh.com (E.S.-K.M.); 2Research Department, Hong Kong Sanatorium and Hospital, Hong Kong Special Administrative Region, Hong Kong 999077, China; linghiu.fung@hksh.com (T.L.-H.F.); wpluk@hksh.com (L.W.-P.L.); 3Hospital Administration, Hong Kong Sanatorium and Hospital, Hong Kong Special Administrative Region, Hong Kong 999077, China; chorchiu.lau@hksh.com

**Keywords:** COVID-19, heterologous vaccination, booster, inactivated virus vaccine, mRNA vaccine, variants, neutralizing antibody

## Abstract

COVID-19 has swept across the globe since 2019 and repeated waves of infection have been caused by different variants of the original SARS-CoV-2 (wild type), with the Omicron and Delta variants having dominated recently. Vaccination is among the most important measures in the absence of widespread use of antivirals for prevention of morbidity and mortality. Inactivated virus vaccine has been abundantly used in many countries as the primary two-dose regimen. We aim to study the safety and immunogenicity of CoronaVac (three-dose inactivated virus vaccine) and the BNT162b2 (two-dose inactivated virus vaccine followed by an mRNA vaccine) booster. Both CoronaVac and BNT162b2 boosters are generally safe and have good immunogenicity against the wild type SARS-CoV-2 and the Delta variant with the majority having neutralizing antibodies (NAb) on day 30 and day 90. However, the BNT162b2 booster is associated with a much higher proportion of positive NAb against the Omicron variant. Only 8% of day 30 and day 90 samples post CoronaVac booster have NAb against the Omicron variant. In addition, more BNT162b2 booster recipients are having positive T-cell responses using interferon gamma release assay. In places using inactivated virus vaccine as the primary two-dose scheme, the heterologous mRNA vaccine booster is safe and more immunogenic against the Omicron variant and should be considered as a preferred option during the current outbreak.

## 1. Introduction

Following the emergence of SARS-CoV-2 in China in late 2019, Hong Kong (HK) was among the first few places to which COVID-19 spread before the rest of the world was affected [1]. The first confirmed case was found on 23 Jan 2020 and, two years down the road, this city still has the lowest incidence rate and mortality rate in the world [2,3]. The majority of the population is, therefore, not immune to the virus, and vaccination becomes the most important measure for prevention of infection and in particular morbidity and mortality from COVID-19 infection. Unfortunately, starting from February 2022, the city was hit by a major outbreak due to the Omicron variant and more than 1 million people were infected [3].

Two vaccines of different platforms are available in HK. The inactivated virus vaccine CoronaVac (Sinovac Biotech Co., Ltd., Beijing, China) and the messenger RNA-based vaccine BNT162b2 (Pfizer-BioNTech Inc., New York, NY, USA) were made available from 26 February 2021 and 10 March 2021, respectively. The inactivated virus vaccine BBIBP CoR V (Sinopharm/Beijing Institute of Biological Products, Beijing, China) was available in China and some of the HK residents had received them in China. Both CoronaVac and BBIBP-CoR V are grown from Vero cell cultures, inactivated by β-Propiolactone and adjuvanted with aluminum. The CoronaVac contains 3 µg virion and the BBIBP-CoR V contains 4 µg total protein. Eleven months after the initiation of the vaccination program, only 74.3% of the population aged 12 or older are vaccinated with two doses (62% with BNT162b2 and 38% with CoronaVac), and on 11 November 2021, a city-wide third dose booster program was initiated and 18.8% of residents received a third dose booster by mid-February 2022 [4]. The unexpected arrival of the Omicron variant has swept across the globe and HK has been hit by both the Delta and Omicron variants at the time of writing. Therefore, this study aims to understand the safety and the potential protection offered by different kinds of boosters and to formulate a strategy that is most suitable for places where different platforms of vaccines are used.

## 2. Materials and Methods

### 2.1. Study Design and Participants

This was an open label trial completed in two centers, the Hong Kong Sanatorium and Hospital (HKSH) and the HKSH Eastern Medical Centre in Hong Kong. A total of 376 individuals completed with two doses of COVID vaccine (CoronaVac, n = 241; BNT162b2, n = 99; BBIBP-CoR V, n = 36) were screened for the neutralizing antibody (NAb). Blood samples were taken at least 90 days after the second dose. Neutralizing antibody (NAb) was done by a surrogate virus neutralization test (SVNT) called GenScript c-Pass SARS-CoV-2 Neutralization Antibody Detection Kit. A total of 240 participants were negative for NAb and were therefore eligible. Six were excluded as they received the mRNA vaccine for the first two doses. The remaining 234 participants could choose either the inactivated virus vaccine CoronaVac (Sinovac Biotech, Beijing, China) or the messenger RNA-based vaccine BNT162b2 (Pfizer-BioNTech Inc., New York, NY, USA) as a third dose booster. The median time interval between the second and third dose was 213 days (range 205–265 days) for BBIBP-CoR and 202 days (94–222 days) for CoronaVac recipients respectively. Blood samples were taken before the third dose booster and at day 30 and 90 for immunogenicity study. Twelve and a further 19 participants failed to provide blood samples at day 30 and 90 respectively due to loss of follow up. An electronic questionnaire regarding post vaccination adverse effects was sent by SMS to every participant weekly for four weeks after the booster for studying of the safety of the booster. All participants were well informed about the study design, and signed informed consent was obtained. The study was approved by the Research Ethics Committee of the HKSH Medical Group (REC-2021-14). All of the collected samples were analyzed for IgG against SARS-CoV-2 Spike protein and NAb, whereas 178 baseline and day-30 samples were tested for T cell-mediated immune response.

### 2.2. Automated Chemiluminescent Anti-Spike IgG Assay

A fully automated platform used in this study was the Abbott Alinity SARS-CoV-2 IgG II Quant assay (Abbott Diagnostics, North Chicago, IL, USA). It is a two-step chemiluminescent microparticle immunoassay (CMIA) used for the quantitative determination of anti-spike protein IgG antibodies to SARS-CoV-2 in human serum or plasma on the Alinity i immunoassay analyzer. Samples and the SARS-CoV-2 recombinant spike protein antigen coated paramagnetic microparticles were mixed in the first incubation. The SARS-CoV-2 IgG antibodies in the sample bound to the SARS-CoV-2 antigen coated microparticles. After washing, anti-human IgG acridinium-labeled conjugate was added to form complexes with the SARS-CoV-2 IgG bound microparticles in the second incubation. Following a wash step, the pre-trigger and trigger substrate solutions were added to create a chemiluminescent reaction which was measured as RLU. The amount of SARS-CoV-2 IgG in the sample was directly proportional to the RLU detected by the Alinity i analyzer. The manufacturer’s recommended positive concentration was ≥7.1 BAU/mL. The World Health Organization (WHO) has established an international standard and reference panel for anti-SARS-CoV-2 antibody for easier comparison between different laboratories using different platforms. The binding antibody unit per mL (BAU/mL) is used as the recommended unit and 1 AU/mL is equivalent to 0.142 BAU/mL by the above method.

### 2.3. Surrogate Neutralizing Antibody Immunoassay

A GenScript cPassTM SARS-CoV-2 Neutralization Antibody Detection Kit (GenScript Biotech, Piscataway, NJ, USA) with three different target of recombinant receptor binding domain (RBD) conjugate was used to determine the NAb level against Wild Type (WT), Delta (L452R), and Omicron variant. GenScript ELISA were performed according to the manufacturer’s instructions. Serum samples and controls were 1:10 diluted in sample buffer and incubated with equal volume of horseradish peroxidase-conjugated receptor binding domain (HRP-RBD) at 37 °C for 30 min. Next, the mixture was added to the recombinant human angiotensin converting enzyme-2 receptor (ACE2) pre-coated plates and incubated at 37 °C for 15 min. The presence of SARS-CoV-2 NAb in the sample bound to the HRP-RBD to form complexes that inhibited the HRP-RBD from binding to ACE2. After washing with buffer, only the non-NAb-bound HRP-RBD remained attached to ACE2. Upon the addition of the chromogenic substrate, color change occurred during the 15 min of incubation at room temperature. A stop solution was added to stop the reaction before reading the absorbance by spectrophotometry at 450nm by microplate reader. The intensity of color formed was inversely proportional to the concentration of NAb in the sample. The obtained optical density (OD) values were used to calculate the percentage of inhibition as (1–sample OD/negative control OD) × 100%. Results ≥30% inhibition were interpreted as positive.

### 2.4. T cell-Mediated Immune Response Assay

A QuantiFERON^®^ SARS-CoV-2 Starter kit (Qiagen, Germantown, WI, USA) was used for the detection of a T cell-mediated immune response. Together with negative and positive control tubes, the starter set also comprises a combination of blood collection tubes containing the original SARS-CoV-2 spike peptides formulation (Ag1 and Ag2) to stimulate lymphocytes in heparinized whole blood involved in cell-mediated immunity. The SARS-CoV-2 Ag 1 tube contained CD4+ epitopes derived from the S1 subunit RDB of spike protein, whereas Ag2 tube contained CD4+ and CD8+ epitopes from the S1 and S2 subunit of the spike protein. After blood collection, each set of blood tubes were mixed thoroughly and incubated at 37 °C for 16–24 h according to the manufacturer’s instruction. In this study, 20 h of incubation was carried out on each set of blood samples. Then the plasma from the stimulated samples was used for the detection of interferon gamma (IFN-ɣ) by QuantiFERON^®^ ELISA with a Dynex^®^ DS2 ELISA processing system. The quoted reference cutoff of interferon gamma level ≥0.15 IU/mL was considered as positive. [5,6,7,8]

### 2.5. Statistics

In this study, two sample t-tests and Fisher’s Exact test were used to calculate the *p*-values for variables in demographic characteristics, pre-existing co-morbidities and adverse effects. The scatter plot graphs of different SARS-CoV-2 antibody response tests and their statistical significance in level comparison were computed by a Mann-Whitney test using Prism version 9 GraphPad software. Computation was done using R version 4.1.0 [9].

## 3. Results

### 3.1. Demographics and Co-Morbidities

A total of 240 participants were recruited, with 98 choosing CoronaVac and 136 choosing BNT162b2 as the third dose booster, respectively (Figure 1). Six were excluded as they have BNT162b2 for the first two doses. One hundred and ninety eight had CoronaVac (84.6%) and 36 had BBIBP-CoR V (15.4%). Table 1 shows participants choosing BNT162b2 as the booster were older than the CoronaVac group (57.9 vs. 54.0 *p* = 0.0111). There was no difference in gender but more participants with a history of cancer were in the BNT162b2 booster group. More diabetic elderly patients (>65) chose CoronaVac (40%) than BNT162b2 (11.6%, *p* = 0.017).

### 3.2. Immunogenicity of Vaccines Determined by Anti-Spike RBD IgG

The baseline anti-Spike RBD IgG was slightly higher in the CoronaVac group, but the NAb between the two groups were not statistically significant. A total of 222 and 203 participants had their blood samples taken on post booster dose day 30 and 90, respectively. The 30-day and 90-day median IgG against the Spike protein RBD was much higher in the BNT162b2 booster group (2302 BAU/mL vs. 143 BAU/mL for day-30, *p* < 0.0001 and 968 BAU/mL vs. 86 BAU/mL for day-90, *p* < 0.0001) than the CoronaVac booster (Table 2 and Figure 2). Both groups showed waning of the IgG level from day 30 to day 90, with BNT162b2 booster group (−57.9%) having more pronounced decline than the CoronaVac group (−39.8%). It was also presented in our previous study that the IgG level declined more in the BNT162b2 recipients than the CoronaVac recipients for the first two doses [10]. Subgroup analysis shows that those with BBIBP-CoR V as the primary vaccination had a lower anti-Spike RBD IgG at baseline, day 30 and day 90 compared to those having CoronaVac as primary two-dose vaccine in both CoronaVac and BNT162b2 groups (Appendix A).

### 3.3. Immunogenicity of Vaccines Determined by Surrogate Neutralizing Antibody Immunoassay

The NAb against the WT, the Delta variant and the Omicron variant were tested by surrogate SARS-CoV-2 neutralizing antibody immunoassays for both groups on day 30 and day 90 post booster vaccination. The BNT162b2 group had a statistically significant higher percentage of positive NAb against WT (100%, 130/130 vs. 90%, 83/92 for day-30, *p* < 0.0001; 99%, 125/126 vs. 79%, 61/77 for day-90, *p* < 0.0001), the Delta variant (99%, 129/130 vs. 82%, 75/92 for day-30, *p* < 0.0001; 97%, 122/126 vs. 70%, 54/77 for day-90, *p* < 0.0001) and the Omicron variant (69%, 90/130 vs. 7%, 6/92 for day-30, *p* < 0.0001; 48%, 61/126 vs. 6%, 5/77 for day-90, *p* < 0.0001) than the CoronaVac group. Although the CoronaVac booster elicits positive NAb against the Delta variant in the majority (82% for day-30 and 70 % on day-90), only 8% of the participants in this group had positive NAb against the Omicron variant on day 30 and day 90 post booster vaccination (Table 2 and Figure 3). Subgroup analysis showed that those with BBIBP-CoR V as primary vaccination have fewer positive NAb compared to CoronaVac primary vaccinated recipients against the WT (75%, 9/12 vs. 93%, 74/80 for day-30, *p* = 0.0913; 50%, 5/10 vs. 84%, 56/67 for day-90, *p* = 0.0279) and Delta variant (33%, 4/12 vs. 89%, 71/80 for day-30, *p* < 0.0001; 30%, 3/10 vs. 76%, 51/67 for day-90, *p* = 0.0062) in the CoronaVac group, but no statistical difference was found against the Omicron variant (8%, 1/12 vs. 8%, 6/80 for day-30, *p* = 1; 10%, 1/10 vs. 7%, 5/67 for day-90, *p* = 1). In the BNT162b2 group, both BBIBP-CoR V and CoronaVac primary vaccination recipients had a high percentage of NAb against WT (100%, 20/20 vs. 100%, 110/110 for day-30, *p* = 1; 94%, 16/17 vs. 100%, 109/109 for day-90, *p* = 0.1349) and Delta (95%, 19/20 vs. 100%, 110/110 for day-30, *p* = 0.1538; 94%, 16/17 vs. 97%, 106/109 for day-90, *p* = 0.4442). However, there was a statistically significantly higher percentage of positive NAb against the Omicron variant for those primarily vaccinated with CoronaVac compared to BBIBP-CoR V primary vaccinated (72%, 79/100 vs. 55%, 11/20 at day 30, *p* = 0.1865; 52%, 57/109 vs. 24%, 4/17 for day-90, *p* = 0.0363) (Appendix A). The median percentage inhibition of the NAb declined with time against WT and the variants. For the CoronaVac group, the decline rate from day 30 to day 90 for the WT and the Delta variant was −23.5% and −36.4% respectively. The decline of the median percentage inhibition of NAb against the Omicron variant in the CoronaVac group is not calculated, as most of the samples showed a low level of inhibition on both day 30 and day 90 (only 8% were positive for NAb) and the fluctuation makes meaningful calculation impossible. For the BNT162b2 group, there is no decline in the median percentage inhibition of NAb against the WT and Delta variant, but there was a −48.4% decline against the Omicron variant between day 30 and day 90. The median fold difference of the percentage inhibition of the NAb between the Delta variant and the Omicron Variant for the CoronaVac group were 35.5 at day 30 and 7.1 at day 90 respectively, whereas for the BNT162b2 group, the median fold difference were 1.8 at day 30 and 3.4 at day 90, respectively (Table 3).

### 3.4. Immunogenicity of Vaccines Determined by T-cell Mediated Immune Response Assay

Blood samples were taken before and 30 days after the third dose booster for the interferon-gamma release assay (IGRA) against the SARS-CoV-2 spike protein as a surrogate for testing T-cell cell mediated immune response. A total of 178 participants (74 from CoronaVac and 104 from BNT162b2) had been tested. After the standard two-dose vaccination, a comparable proportion of participants from the CoronaVac group and from the BNT162b2 group had a positive interferon gamma release response (33.8%, 25/74 vs. 30.8%, 32/104, *p* = 0.96). A significantly higher proportion of the participants from the BNT162b2 booster group became positive for IGRA than the CoronaVac booster group (86.5%, 90/104 vs. 63.5%, 47/74 *p* < 0.0001) (Table 4 and Figure 4).

### 3.5. Safety

Seventeen participants (7.3%) failed to respond to our electronic questionnaire. Statistically more participants from the BNT162b2 group had adverse effects than the CoronaVac group (50.7%, 69/136 vs. 16.3%, 16/98, *p* < 0.0001). The onset of the adverse effects was within the first week in both groups, but fever, injection site pain, fatigue, myalgia, arthralgia and nasal congestion were much more prevalent in the BNT162b2 group (Table 5). All the reported side effects were mild to moderate in degree. No participants suffered from any serious adverse effects and both platforms appeared to be safe as a booster.

## 4. Discussion

In this study, we aimed to understand the safety and immunogenicity of the CoronaVac and BNT162b2 booster after a primary two-dose inactivated virus vaccination against the current circulating variants of concern, namely Omicron and Delta. Both the CoronaVac and BNT162b2 booster vaccinations are safe. The mRNA vaccine booster is associated with more adverse effects, but they are mostly mild and self-limiting. The same kind of pattern is also evident in other studies [11,12]. In fact, our cohort shows less injection site pain (39.5% vs. 76–90%), comparable myalgia (34.9% vs. 23–56%), but more fever (14.7% vs. 1–2%) [11,12].

Although this study did not exactly compare a homologous vs. heterologous approach of booster vaccination, as some participants had BBIBP-CoR V in the primary vaccination regime and received CoronaVac as a booster, it is still comparing the prime-boost vaccination using a single platform (i.e., the inactivated virus vaccine, BBIBP-CoR V and CoronaVac) versus a heterologous approach using a mRNA vaccine (BNT162b2) as a booster. The immunogenicity is stronger for the BNT162b2 group compared to the CoronaVac group. The BBIBP-CoR V subgroup had lower IgG and NAb positivity (for WT and Delta in the CoronaVac booster group and for Omicron in the BNT162b2 booster group), compared to the CoronaVac subgroup. Those received BBIBP-CoR V as the primary vaccination regime had a significantly longer interval between the second and third dose vaccine (Appendix A). NAb has been advocated to be one of the most important markers for protection against COVID-19. Previous studies have shown that a heterologous mRNA booster is more immunogenic than a homologous booster in two dose inactivated virus primary vaccination [11,12,13,14]. Zhang et al has shown a 6.4 fold increase in NAb after a heterologous mRNA vaccine booster compared to a homologous booster in mouse model at day 14 [13]. In a large Brazilian study [12], participants were randomized to receive an mRNA booster, one of the two recombinant adenoviral vectored vaccine boosters, or the homologous inactivated virus vaccine booster. The anti-Spike IgG and pseudovirus NAb (against WT) geometric mean titre of the heterologous mRNA booster group were found to be 13.4 fold and 21.5 fold higher than the homologous inactivated virus vaccine group at day 28. A small subgroup was subjected to live virus neutralization against the Delta variant, and 80% of the homologous and 100% of the heterologous booster recipients are able to mount NAb [12]. When compared to the homologous approach, a recent Dominican Republic study [14] has also shown that a heterologous mRNA booster in primary two-dose inactivated virus vaccine recipients elicited 13.4-fold anti-Spike IgG, 10.1- and 6.3-fold NAb against WT, and the Delta variant, respectively. Several studies have shown that a heterologous prime boost vaccination using the adenovirus-based vector vaccine (ChAdOx1) followed by the mRNA vaccine such as BNT162b2/mRNA-1273 had a higher NAb as well as stronger T-cells immune stimulation too [15,16,17].

As the Omicron variant has significant changes in the Spike protein due to the many mutations [18], it is expected that the currently available vaccines may not be as effective when compared to WT or earlier variants. Data on NAb against the Omicron variant after the homologous and heterologous approach is scarce, and most of the existing data were short term (only one month post booster). The persistence of such NAb is yet to be known. The Dominican Republic study [17] showed that 80% of the recipients have NAb against the Omicron variant 28 days after the heterologous mRNA vaccine booster. Cheng et al [19] also showed that 80% of the heterologous recipients and 3% of the homologous booster generated NAb against the Omicron variant three to five weeks post-booster. In our study, the heterologous mRNA booster enables a reasonably high proportion of recipients (69% at 1-month and 48% at 3-month post booster) to produce NAb against the Omicron variant. Our result also echoed the failure of the homologous booster to elicit NAb against the Omicron variant in the vast majority (93%). This is unlikely to be effective in controlling the spread of the infection, particularly for the very transmissible Omicron variant, if the homologous approach is adopted. In a Syrian hamster study, both weak and potent NAb were shown to be effective in preventing infection [20]. Clinical studies have also shown that, after infection with SARS-CoV-2, the higher the level of NAb, the less likely that the patient would become infectious [21]. It is therefore important to consider these factors in defining the strategy of which platform to use as the booster.

Cell mediated immunity is an equally important element of the immune system to handle COVID infection, both for limiting the extent of disease as well as for the prevention of initial infection [22]. Data on T cell immunity after both the homologous and heterologous approaches are lacking. The T cell immunity is shown to have a better boosting effect by the heterologous mRNA booster in a mice model, with a 4.83-fold increase in interferon-gamma release after being stimulated by the S1-RBD peptide [16]. A study has shown a positive relation between early detection of interferon-gamma secreting T-cells and the control of SARS-CoV-2 infection [23]. It is therefore vital that the booster can stimulate the cell-mediated immunity. We have shown that the heterologous mRNA booster resulted in stronger interferon gamma release after spike protein peptide stimulation at day 30. Kanokudom et al has also shown similar findings at day 14 and 28 post booster [14].

There are several limitations in this non-randomized study. First, the small sample size with all the recruited participants had inactivated vaccines as their first two doses. Most mRNA vaccine recipients (93 out of 99) were still having NAb at the time of recruitment and therefore were not eligible. Secondly, we have only used surrogate SARS-CoV-2 NAb immunoassays but not live virus plaque reduction assays to study the B-cell response against the WT, the Delta variant and the Omicron variant. Studies have shown comparable and reliable results using surrogate virus assays, and it avoids the need for the much higher biosafety level in handling the live virus [24]. In addition, we have only used a commercially available IGRA to study the T-cell response before and after the third dose booster. This particular version used two peptides from the spike protein to trigger the T-cell response to release the IFN-ɣ. Therefore, T-cell responses from other antigens (membrane protein or nucleocapsid protein) are not investigated. Inactivated vaccines, containing the whole virion, may have better T-cell responses if the tests include these proteins in addition to the spike protein, though studies have shown responses to the spike protein to be the most abundant [25].

## 5. Conclusions

In this study, we have shown both CoronaVac and the BNT162b2 booster in primary inactivated virus vaccine recipients are safe and immunogenic. A heterologous booster with BNT162b2 has a much better NAb and T cell response against the circulating Omicron variant. The effect still lasts up to 90 days post booster vaccination.

## Figures and Tables

**Figure 1 vaccines-10-00556-f001:**
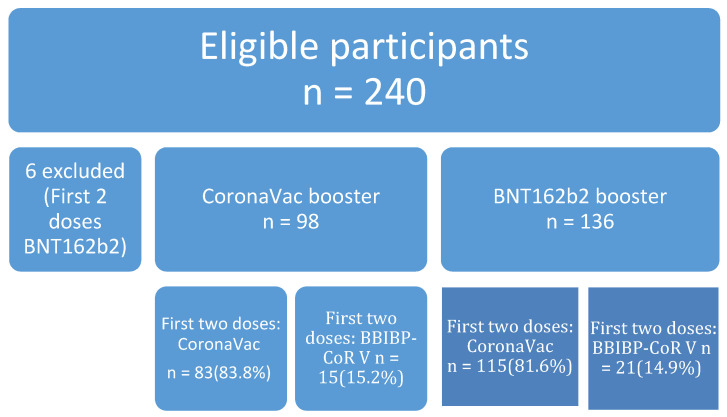
Participant’s recruitment and their corresponding primary vaccination status.

**Figure 2 vaccines-10-00556-f002:**
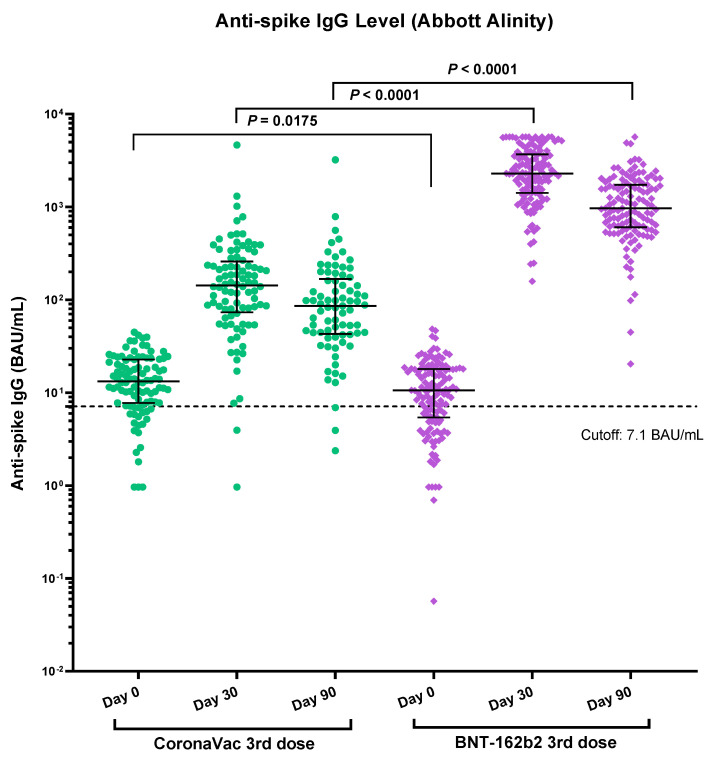
Comparison of immunogenicity of CoronaVac and BNT162b2 booster vaccination by quantitative anti-spike IgG. Number of samples at day 0, day 30 and day 90 of CoronaVac (N = 98, 92, 77) and BNT162b2 (N = 136, 130, 126), respectively. Median and *p*-values were tested using a Mann-Whitney test (By Prism GraphPad Software).

**Figure 3 vaccines-10-00556-f003:**
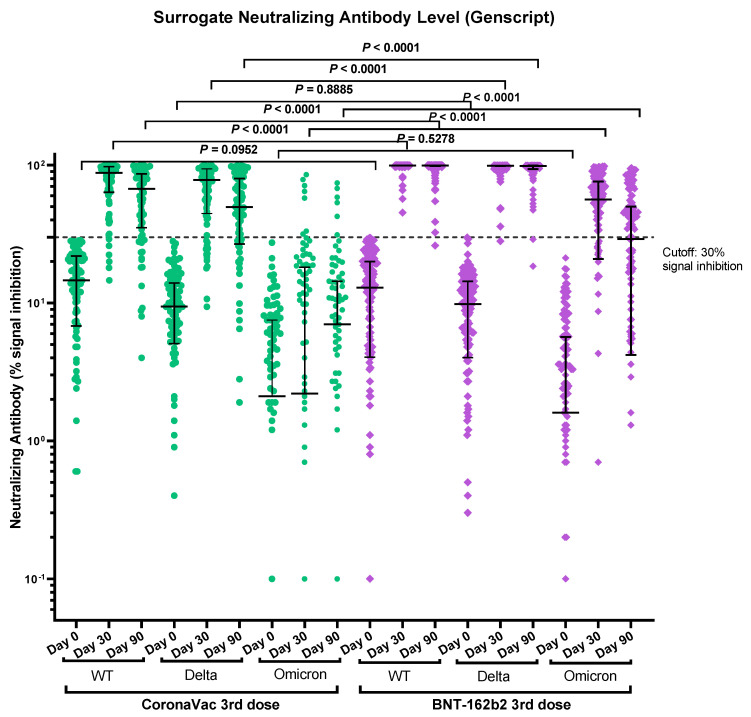
Comparison of immunogenicity of CoronaVac and BNT162b2 booster vaccination by surrogate NAb assay. Median and *p*-values were tested using a Mann-Whitney test (By Prism GraphPad Software).

**Figure 4 vaccines-10-00556-f004:**
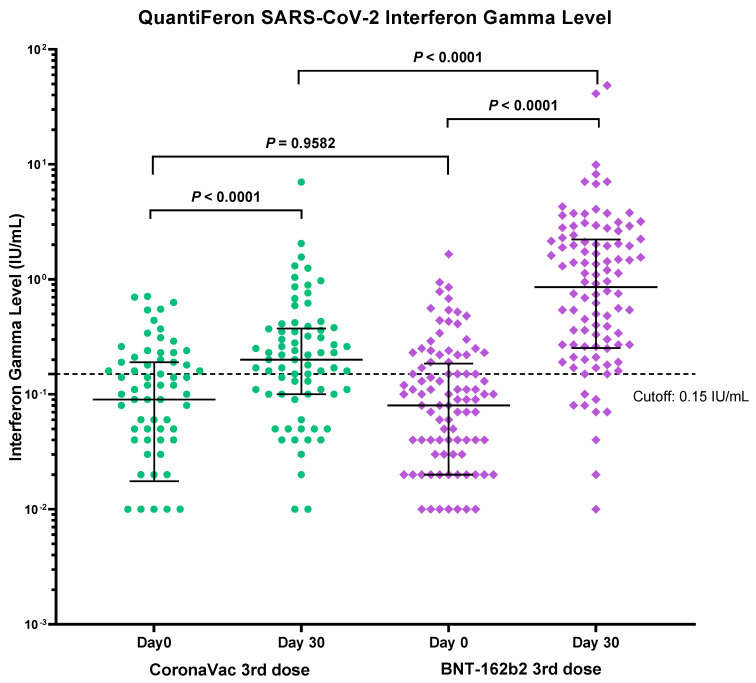
Comparison of immunogenicity of CoronaVac and BNT162b2 booster vaccination by QuantiFERON SARS-CoV-2 (IGRA). Median and *p*-values were tested using a Mann-Whitney test (By Prism GraphPad Software).

**Table 1 vaccines-10-00556-t001:** Demographic characteristics and pre-existing co-morbidities **^1^**.

	CoronaVac Group (n = 98)		BNT162b2 Group (n = 136)		*p*-Value ^2^ (<65)	*p*-Value ^2^ (≥65)
	Age < 65 (n = 78)	Age ≥ 65 (n = 20)	Age < 65 (n = 93)	Age ≥ 65 (n = 43)		
Female	42 (53.8%)	5 (25%)	53 (57.0%)	12 (27.9%)	0.76	>0.99
Mean age	49.9 (S.D.9.0)	70.4 (S.D.3.4)	52.0 (S.D.7.8)	70.9 (S.D.5.6)	0.10	0.65
Cardiovascular diseases	1 (1.3%)	6 (30.0%)	1 (1.1%)	6 (14.0%)	>0.99	0.17
Stroke	0	0	1 (1.1%)	0	>0.99	>0.99
Hypertension	16 (20.5%)	10 (50.0%)	19 (20.4%)	20 (46.5%)	>0.99	>0.99
Diabetes mellitus	4 (5.1%)	8 (40.0%)	6 (6.5%)	5 (11.6%)	0.76	0.017
Hyperlipidemia	14 (17.9%)	6 (30.0%)	12 (12.9%)	12 (27.9%)	0.40	>0.99
Asthma	2 (2.6%)	0	0	2 (4.7%)	0.21	>0.99
Chronic renal diseases	0	0	1 (1.1%)	0	>0.99	>0.99
Chronic liver diseases	2 (2.6%)	0	1 (1.1%)	0	0.59	>0.99
Cancer	1 (1.3%)	0	7 (7.5%)	7 (16.3%)	0.07	>0.99
Received chemotherapy and/or radiotherapy	0	0	0	0	>0.99	>0.99
Steroid ^3^	0	1 (5.0%)	3 (3.2%)	2 (4.7%)	0.25	>0.99
AIDS ^4^	0	0	0	0	>0.99	>0.99
Systemic lupus erythematosus	0	0	1 (1.1%)	0	>0.99	>0.99
Other autoimmune diseases	1 (1.3%)	0	6 (6.5%)	2 (4.7%)	0.12	>0.99
Food/drug allergy	0	0	0	0	>0.99	>0.99

^1^ Individual may have more than one co-morbidity. ^2^ *p* < 0.05, the results are significantly different. ^3^ Steroid dosage of ≥7.5 mg per day prednisolone equivalents. ^4^ AIDS: Acquired Immunodeficiency (Virus) Syndrome. Mean age was tested using two sample t-test. Other variables were tested using Fisher’s exact test.

**Table 2 vaccines-10-00556-t002:** Quantitative IgG level against Spike protein and neutralizing antibody (NAb) positivity after CoronaVac and BNT162b2 booster.

		CoronaVac Group (n = 98)	BNT162b2 Group (n = 136)	*p*-Value
Antibody level, BAU/mL, median (IQR *)	Baseline	13.2 (7.8–22.8) n = 98	10.6 (5.4–18.0) n = 136	<0.0175
	Day 30	143 (73.4–259) n = 92	2302 (1414–3677) n = 130	<0.0001
	Day 90	86.1 (43.0–167.8) n = 77	968 (604.2–1740) n = 126	<0.0001
NAb positivity against wild type	Day 30	90% (83/92)	100% (130/130)	0.0003
	Day 90	79% (61/77)	99% (125/126)	<0.0001
NAb positivity against Delta	Day 30	82% (75/92)	99% (129/130)	<0.0001
	Day 90	70% (54/77)	97% (122/126)	<0.0001
NAb positivity against Omicron	Day 30	7% (6/92)	69% (90/130)	<0.0001
	Day 90	6% (5/77)	48% (61/126)	<0.0001

* Interquartile range. Median and *p*-values were tested using Mann-Whitney test (By Prism GraphPad Software).

**Table 3 vaccines-10-00556-t003:** Decline rate of the NAb percentage inhibition against the WT, Delta and Omicron variant over time and Median Fold Change of percentage inhibition of NAb on day 30 and day 90 between the Delta and Omicron variants.

		CoronaVac Group (n = 98)	Decline Rate	Fold Change	BNT162b2 Group (n = 136)	Decline Rate	Fold Change
NAb Median percentage inhibition against WT	Day 30	87.9 (63.6–97.7) n = 92	Day 30 to 90: −23.5%		99.5 (99.3–99.6) n = 130	Day 30 to 90: 0.0%	
	Day 90	67.2 (35.2–86.5) n = 77			99.5 (98.7–99.6) n = 126		
NAb Median percentage inhibition against Delta	Day 30	78.2 (44.6–94.1) n = 92	Day 30 to 90: −36.4%	Delta Day 30 vs. Omicron Day 30: 35.5	99.1 (98.3–99.3) n = 130	Day 30 to 90: −0.3%	Delta Day 30 vs. Omicron Day 30: 1.8
	Day 90	49.7 (26.8–80.2) n = 77			98.8 (93.6–99.4) n = 126		
NAb Median percentage inhibition against Omicron	Day 30	2.2 (0–18.2) n = 92	Day 30 to 90: 218.2% *	Delta Day 90 vs. Omicron Day 90: 7.1	56.4 (20.9–75.9) n = 130	Day 30 to 90: −48.4%	Delta Day 90 vs. Omicron Day 90: 3.4
	Day 90	7 (0–14.4) n = 77			29.1 (4.2–50.1) n = 126		

* most samples are negative for NAb and the median percentage inhibition is not representative of the natural decline. Median and *p*-values were tested using a Mann-Whitney test (By Prism GraphPad Software).

**Table 4 vaccines-10-00556-t004:** QuantiFERON-SARS-CoV-2 positivity before and after CoronaVac and BNT162b2 booster.

	CoronaVac Group (n = 74)	BNT162b2 Group (n = 104)	*p*-Value ^1^
Positive at baseline	33.8 % (25/74)	30.8% (32/104)	*p* = 0.9582
Positive at day 30	63.5% (47/74)	86.5% (90/104)	*p* < 0.0001

^1^ *p*-values were tested using Mann-Whitney test.

**Table 5 vaccines-10-00556-t005:** Adverse effects of CoronaVac and the BNT162b2 booster **^1^**.

	CoronaVac Group (n = 98)	BNT162b2 Group (n = 136)	*p*-Value
No data	10.2% (10/98)	5.0% (7/136)	
No adverse effect	74.5% (73/98)	47.8% (65/136)	
Any adverse effect	16.3% (16/98)	50.7% (69/136)	<0.0001
Fever	4.5% (4/88)	14.7% (19/129)	0.0230
Chills	8.0% (7/88)	7.0% (9/129)	0.7967
Injection site pain	13.6% (12/88)	39.5% (51/129)	<0.0001
Fatigue	6.8% (6/88)	32.6% (42/129)	<0.0001
Headache	8.0% (7/88)	14.0% (18/129)	0.1995
Muscle pain	2.3% (2/88)	34.9% (45/129)	<0.0001
Diarrhoea	2.3% (2/88)	5.4% (7/129)	0.3170
Joint pain	1.1% (1/88)	14.7% (19/129)	0.0005
Skin rash	2.3% (2/88)	3.9% (5/129)	0.7037
Nausea/vomiting	1.1% (1/88)	2.3% (3/129)	0.6483
Tremor	0	1.6% (2/129)	0.5156
Abdominal pain	0	0.8% (1/129)	>0.99
Urticaria	0	1.6% (2/129)	0.5156
Enlarged lymph nodes	0	1.6% (2/129)	0.5156
Sore throat	0	3.9% (5/129)	0.0821
Nasal congestion	0	5.4% (7/129)	0.0432
SAE ^1^	0	0	>0.99

^1^ Serious Adverse Events were defined as vaccine-related undesired events, including disability, life-threatening conditions and death. All variables were tested using Fisher’s exact test.

## Data Availability

The data used to support the findings of this study are included within the article.

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
