# Peer review of "Safety and Immunogenicity of a Booster Vaccination by CoronaVac or BNT162b2 in Previously Two-Dose Inactivated Virus Vaccinated Individuals with Negative Neutralizing Antibody"

_vaccines, 2022, doi:10.3390/vaccines10040556_

Round 1

Reviewer 1 Report

Minor comments:

  1. Authors mentioned that 15 patients from CoronaVac booster group and 21 subjects from BNT162b2 booster group had received BBIBP-CoR vaccine while others received CoronaVac vaccine. There is no analysis whether the type of inactivated virus vaccine has any infuence on the outcome of different boosters. Although differences are unlikely it will be useful to add such analysis.
  2. It is important to mention in Conlcusions that measurements were performed up to 90 days post booster which is novel and important for the field.

Author Response

  1. Thank you for the suggestion. A subgroup analysis comparing those received BBIBP-CoR V with CoronaVac was done. Results are incorporated into the manuscript (Line 172-176; Line 199-213).
  2. It is incorporated into conclusion (Line 356).

Reviewer 2 Report

The manuscript is clear and well written. The cited references are appropriate and  current and does not include self citation. The study aimed to compare the safety and immunogenicity of a heterologous booster (BNT162b2) with a homologous booster (Corona Vac) in adults who received 2 primary doses of Corona Vac. It is crucial to recommend a safe and effective booster vaccination against the recent variants (Delta and Omicron) in order to protect the population in Hong Kong during this current COVID-19 outbreak.

In the Introduction, the authors should mention the date of initiation of COVID-booster in Hong Kong. The study design was open label and completed in 2 centres. The study design could have been improved by converting it to participant blinded, randomized trial to eliminate potential biasedness. The exclusion criteria and the interval between the second dose of Corona Vac and the third dose (booster) should be specified in the Method. The participants were followed up to 90 days post booster, which is longer than the other booster studies. Safety data was collected by electronic questionnaire regarding post vaccination adverse events was collected for 4 weeks after the booster. The extended period for safety and immunogenicity provided more information on the safety and long-term protection against SARS-CoV-2.

In the results (Table 1), the number of elderly subjects, 65 years and older, should be differentiated from the younger adults. The immune responses to booster doses in the elderly is important as they are a vulnerable group with high mortality. and morbidity. Regarding antibody assay, testing for SARS-CoV-2 nucleocapsid IgG should be included. This will provided additional information about previous infection with SARS-CoV-2 prior to administering the booster.  Interferon-gamma release assay against SARS-CoV-2 spike protein was tested as a surrogate for T-cell cell mediated immune response. The authors recognized the limitation of this assay in investigating the T-cell responses elicited by COVID-19 vaccines.

The figures and tables in the manuscript are appropriate and show the results properly. The data is interpreted appropriately and consistently throughout the manuscript with p values for the different variables in Table 1 and Table 4. The scatter plot graphs of different SARS-C0V-2 antibody response and their statistical significance were computed accurately. The adverse events for the 2 boosters could have been better illustrated in the form of bar graph rather than tabulated (Table 4).

Previous studies have shown that heterologous RNA booster is more immunogenic then homologous booster in two dose inactivated virus vaccinated recipients, notably in the studies conducted in Thailand and Brazil.

A number of caveats have to be mentioned.  First, the relatively small sample size  limits the statistical power of the results obtained. Multi-centre randomized longitudinal studies with larger sample size will determine whether the data is reliable.  Second, there s a need to determine the optimal prime/boost interval of the participants depending on their age and comorbidities. Third, the authors should have determined the live virus focus reduction neutralization tests (FRNT) in the subjects. However, cPass SARS-CoV-2 neutralization Antibody Detection assay was utilized to determine the Nab against Wild Type, Delta and Omicron variants. The authors need to discuss whether neutralizing antibody thresholds are associated with protection from COVID-19 infection in humans. Quantitative correlates of protection have yet to be determined.

The study provided evidence that in countries where inactivated virus vaccine is used mainly as the primary two-dose schedule (prime), heterologous mRNA vaccine booster is safe and more immunogenic against the Omicron variant  compared to the homologous booster.

Author Response

  1. The date of initiation of COVID-booster is mentioned in Line 54;
  2. The exclusion criteria and the interval between the second and third dose are now included in the 'Method' section (Line 67-74);
  3. As suggested, table 1 is now amended (Line 155);
  4. Testing for SARS-CoV-2 nucleocapsid IgG is not included because of two reasons. Firstly, at the time of recruiting the participants, the city has only had 12000 cases (0.16% of population). All cases have to be notified to the department of health and therefore would have been captured during the recruitment. Secondly, infected individuals might have NAb and would be excluded. Thirdly, the inactivated virus vaccine can cause lead to positive IgG against the nucleocapsid;
  5. The small sample size is addressed (Line 337);
  6. As suggested, a larger study to look for the optimal prime/boost interval is discussed (Line 340-342);
  7. We understand virus neutralization assay is the gold standard for determination of NAbs, however, we do not have a biosafety level 3 laboratory in our facility for cell/ viral culture. cPass™ Neutralization Antibody Detection Kit is an appropriate alternative. It is FDA approved and found to have good correlate with plaque reduction neutralization test (PRNT), focus reduction neutralization assay (FRNT) and pseudotyped lentiviral neutralization assay (doi: 10.1128/JCM.02438-20. ;  doi: 10.3390/diagnostics11122193;  doi: 10.1093/ofid/ofab220);
  8. We acknowledge NAb threshold for protection against infection or severe disease is not well defined and yet, a higher NAb level is found to be associated with better protection against symptomatic infection ( doi: 10.1038/s41591-021-01377-8; doi: 10.1038/s41591-021-01540-1).

Reviewer 3 Report

In this paper, the authors delineate a story about booster shots. The authors describe that for participants in this study who were vaccinated with inactivated virus vaccines for the first two shots, the ones who took mRNA vaccines as boosters had better immunogenic response against Omicron comparing to the ones who took the homologous inactivated virus vaccines as boosters.

This paper is well-organized and clear, with good flow and logic and very important meaning in both scientific field and public health, since we are still in the middle of the pandemic. The discussion part is well written, particularly. I personally like the paper. However, there are still some tiny points that would improve the paper.

  1. Please talk about the similarity and difference of the inactivated vaccines from CoronaVac and BBIBP, since that’s the base of your whole research.
  2. Please also briefly mention the machinery of the inactivated vaccines and mRNA vaccines, so readers will get a better idea on the meaning of “homogenous” and “heterogenous”. A few lines, or even an illustration figure, would be enough.
  3. Please go back and check grammar. The flow of some sentences sounds a bit wired.
  4. Another thing is for line 138-141, some of your numbers are written in Arabic numerals and some numbers are shown in English spelling. Please go over your paper and make sure you keep things consistent.

Author Response

  1. Similarity and difference of CoronaVac and BBIBP-CoR V are discussed (Line 49-52);
  2. The machinery of the two vaccine platforms is discussed briefly (Line 52-56);
  3. Thank you. We have checked again;
  4. Those numbers starting a sentence (i.e. after a full stop) are written in English whereas the rest are in Arabic numerals.